# Energy Action Mechanism of Reinforced Sandstone under Triaxial Cyclic Loading and Unloading

**DOI:** 10.3390/ma16010211

**Published:** 2022-12-26

**Authors:** Shuguang Zhang, Yanmo Li, Juefeng Yang, Yu Song, Li Yang, Jiahao Guo

**Affiliations:** 1School of Civil Engineering, Guilin University of Technology, Guilin 541004, China; 2Guangxi Key Laboratory of Geotechnical Mechanics and Engineering, Guilin University of Technology, Guilin 541004, China; 3College of Foreign Languages, Guilin University of Technology, Guilin 541004, China

**Keywords:** rock mechanics, sandstone, triaxial cyclic loading and unloading, reinforcement, CFRP, energy damage

## Abstract

In underground engineering, reinforcement is a necessary means to ensure the stability of surrounding rock. Due to the stress redistribution caused by excavation disturbances, the reinforced rock mass is frequently subjected to loading and unloading, and its mechanical properties change accordingly. Based on the above engineering practice, using pasted circular CFRP, an approximate simulation of the rock reinforcement effect of bolt and shotcrete support was performed. Triaxial cyclic loading and unloading tests of reinforced sandstone were carried out, and the influence of different reinforcement schemes on the mechanical properties was compared and analyzed. Furthermore, the strengthening mechanism, damage evolution, and energy transformation mechanism of CFRP are discussed. The results showed that the peak strength increased about 14.2% and 23.8% with the two reinforced schemes, and the residual strength increased about 27.3% and 52.8% with the increase in the area reinforced by CFRP. Under the same confining pressure and strain conditions, the characteristic energy density and elastic energy ratio increased with an increase in the reinforcement area, but the damage variable decreased. It is proved that CFRP can improve energy absorption efficiency, enhance the energy storage limit, and reduce dissipation efficiency. By inhibiting the propagation of internal fissures and limiting the energy dissipation during fractures, the rock mass can be restrained and strengthened.

## 1. Introduction

In deep engineering, due to the complex stress environment and the influence of construction, rock mass is usually subjected to cyclic load, which leads to a change in mechanical properties and the emergence of safety risks. From the perspective of energy, the failure process of rock is an irreversible process of energy conversion and dissipation [1,2]. Therefore, studying the energy evolution and failure mechanism of reinforced rock during cyclic loading and unloading is of great significance for improving the anchoring reinforcement mechanism and developing engineering support theory [3,4].

In the current research, carbon fiber-reinforced plastic (CFRP), as an excellent material, has been widely used in strengthening concrete structures. Cao et al. analyzed the deformation behavior of expansion concrete under eccentric loads by comparing the bending load and ultimate load of expansion concrete and non-expansion concrete under a CFRP constraint [5]. Zaki et al. conducted loading experiments on three full-size T-beams reinforced with CFRP plates to analyze the anchoring performance of CFRP bolts and verified that the CFRP bolts effectively solved the problem of premature stripping failures of concrete beams [6]. Through loading experiments on 28 CFRP strips with different angles, Ghoroub et al. proposed an equation for the ultimate bearing capacity of these strips according to the test results and established relevant models [7]. Haddad et al. used CFRP cables of different sections to strengthen concrete slabs and concluded that CFRP cables with a 20° dip angle had the best strength improvement effect [8]. Peng et al. proposed a numerical simulation method by testing CFRP steel tubes with various layers to reveal the compression-torsion properties of these steel tubes on square concrete and verified the test results [9]. Tang et al. characterized the relationship between the number of CFRP layers and the ultimate bearing capacity by controlling variables such. as the width/thickness ratio of stainless steel pipe and the number of CFRP layers and proposed an inner steel pipe model to reveal the axial compression characteristics of CFRP square concrete with a stainless steel outer pipe [10]. Li et al. proposed a modified epoxy resin as the binder of CFRP and concrete and tested the comprehensive strength of the bonding interface and the flexural performance of CFRP-reinforced concrete [11]. Ding et al. established and verified the concentric model of CFRP, concrete, and steel pipe based on the theory of continuum mechanics [12]. Carillo et al. believed that CFRP plates could be used to repair the shear strength of light concrete slabs and conducted comparative tests between ordinary concrete slabs and CFRP-refurbished concrete slabs to analyze the differences in their cracking and failure modes [13]. By conducting impact and shear tests on concrete slabs reinforced with CFRP grids in different schemes, Huang et al. drew the conclusion that the scheme of a vertical layout is more effective than a rectangular layout [14]. Wang et al. conducted a comparative test between ordinary steel bar and CFRP-reinforced coral concrete and established the corresponding calculation formula of the maximum crack width of CFRP coral concrete [15]. Dong et al. believed that CFRP could effectively improve the corrosion resistance of concrete piles by conducting corrosion tests under a high temperature and humidity [16].

Although CFRP has been widely used in concrete reinforcement, there are few studies on the reinforcement of rocks by CFRP. At present, the mainstream anchoring method is mainly reinforced bolt anchoring, which will cause initial damage to the sample and affect the test results. CFPR reinforcement can avoid this problem and approach the stress state of the reinforced surrounding rock. Therefore, we studied the mechanical properties, energy accumulation, and dissipation characteristics of reinforced rock samples under the condition of triaxial cyclic loading and unloading and reveal the strengthening mechanism of CFPR from the perspective of energy.

## 2. Test Part

### 2.1. Experiment Material and Test Equipment

The sandstone used in the test was taken from a deep roadway buried at a depth of approximately 800 m. It was hard in texture, with fine particles and no obvious joints on the surface, and its appearance was grayish white. According to the international standard of rock mechanics test, the sandstone was processed into standard cylindrical specimens with a diameter of 50 mm and a height of 100 mm (Figure 1). An RSM-SY8 wave tachometer was used to measure the wave velocity of the sample, and sandstone samples with similar wave velocity were selected for use.

Ring-bonded CFRP was used to strengthen the specimen [17], which can avoid the initial damage caused by the bolt drilling. The CFRP and impregnated adhesive were produced by Carbone Technology Co., LTD., Tianjing, China. The detailed technical parameters are shown in Table 1.

A triaxial cyclic loading and unloading test was completed on the automatic rock triaxial test system (Figure 2). The equipment was composed of dynamic and static servo hydraulic loading systems with a maximum axial force of 2000 kN, a maximum confining pressure of 60 MPa, and a maximum power water pressure of 60 MPa. The machine can carry out various rock mechanics tests, such as loading and unloading, cyclic loading and unloading, a rheological property test, and so on.

### 2.2. Specimen Preparation

The following were the steps used for the preparation of circumferential reinforced specimens.

(1) The CFRP was cut into strips with length of 170 mm and width of 10 mm, which was convenient for impregnating the adhesive and reducing the influence of stress deviation caused by overlap.

(2) The CFRP coated with impregnated glue was pasted on the cylindrical sample according to the reinforcement scheme. Then, the CFRP was strengthened to the appropriate strength by leaving it at room temperature for 72 h until the impregnating glue finally set.

(3) To compare and analyze the influence of CFRP reinforcement on energy evolution, two reinforcement schemes were designed, as shown in Figure 3.

### 2.3. Test Scheme

The confining pressure was set to 5, 10, 15, and 20 MPa. The triaxial cyclic loading test scheme was carried out by the following steps. Hydrostatic pressure was applied at a loading rate of 0.1 MPa/s, and confining pressure remained unchanged after loading to a preset value. The axial stress was loaded to the set value at a displacement rate of 0.05 mm/min, then unloaded to 1 MPa at a rate of 0.8 MPa/s, and then it continued to load to the next load set value. The displacement increment of each cycle was 0.1 mm before failure and 0.05 mm in the residual stage. The sandstone with different reinforcement schemes adopted the same strain for each cycle to facilitate comparative analysis.

## 3. Analysis of Stress–Strain Curves

According to the test results, the stress–strain curves of triaxial cyclic loading and unloading under different confining pressures are roughly the same. To save space, only the stress–strain curve with a confining pressure of 15 MPa is shown in Figure 4.

According to Figure 4, the curve of annular CFRP-reinforced sandstone is similar to that of the original specimen. In each loading and unloading cycle, the unloading curve is slightly lower than the loading curve. When unloaded to the set value, the strain did not recover to the initial value of the cycle, indicating that both elastic deformation and plastic deformation occurred during the loading process. After unloading, the next level of loading was carried out, and the reloading curve intersected the unloading curve to form a stress–strain closure surface, namely, the hysteresis loop [18].

The area of the hysteresis loop can be represented as the area enclosed by the nth unloading curve and the nth + 1 loading curve, that is, the difference between the input energy and the released elastic energy is the dissipated energy [19]. As the cycle progresses, in turn, the hysteresis circle moves backward continuously, and its area becomes larger and larger, which indicates that the dissipated energy accumulates continuously during the cycle, and the damage degree of the rocks increases continuously. The failure of sandstone is caused by the accumulation of dissipated energy [18].

The peak strength, elastic modulus, and residual strength of sandstone were effectively enhanced by CFRP. According to the cyclic loading and unloading test results, the relationship between the peak strength and the confining pressure of the sandstone under different reinforcement conditions is shown in Figure 5. After linear fitting, the Moore–Coulomb criterion was used to calculate the shear strength parameters.

In order to quantitatively analyze the influence of the reinforcement scheme, the mechanical parameters of the reinforced sandstone were calculated when the confining pressure was 15 MPa, and the calculation results are shown in Table 2.

## 4. Analysis of Energy Evolution Characteristics

### 4.1. Method of Energy Calculation

In the triaxial test, the axial force performs positive work along the positive direction of the compressed rock, resulting in axial deformation and energy accumulation. Confining pressure restricts the volume expansion of the rock samples to perform negative work, and radial deformation dissipates energy. In this study, a fixed confining pressure was adopted, and the variation in the radial deformation energy of rock samples was limited and small; it is not the force that directly leads to the failure of rock samples, so the evolution of radial energy is disregarded [20]. In addition, when the rock changes from the initial state to the hydrostatic pressure state, the energy change caused by the internal force of the sample is small, which is not considered in this study [21].

If a single specimen deforms and fails under load, it is regarded as a closed system, that is, it does not exchange heat with the outside world. Then, according to the first law of thermodynamics:(1)$$U=U^{e}+U^{d}$$

Here, *U* is the total energy produced by the work performed by the external force, *U*^e^ is the elastic energy that can be released, and *U*^d^ is the dissipated energy.

According to the energy density calculation method, the calculation schematic is drawn in Figure 6. In a cycle, the area enclosed by the loading curve and the ε axis can characterize the total input energy density *U*^e^. The unloading curve intersects the ε axis at ε^e^, and the area enclosed by the unloading curve and the ε axis can characterize the released elastic energy *U*^e^. The total energy density (*U*) minus the elastic energy (*U*^e^) is the dissipated energy (*U*^d^), which is the area enclosed by the loading and unloading curves.

The calculation formula of *U*^e^ and *U*^d^ is
(2)$$\begin{array}{l} U_{n}=\int \sigma_{n}d\varepsilon_{n}\\ U_{n}^{e}=\int {\sigma^{\prime }}_{n}d{\varepsilon^{\prime }}_{n}\\ U_{n}^{d}=U_{n}-U_{n}^{e}=\int \sigma_{n}d\varepsilon_{n}-\int {\sigma^{\prime }}_{n}d{\varepsilon^{\prime }}_{n} \end{array}$$
where *σ_n_* and *ε_n_* are the loading stress and strain of the nth cycle; *σ’_n_* and *ε’_n_* are the unloading stress and strain of the nth cycle.

### 4.2. Analysis of Energy Evolution

According to the above calculation method, the energy densities *U*, *U*^e^, and *U*^d^ of reinforced sandstone under cyclic loading and unloading conditions are obtained. The variation curves of stress, strain, and energy density under different reinforcement schemes are shown in Figure 7, Figure 8 and Figure 9.

Under the same confining pressure, the energy evolution law of sandstone with different reinforcement schemes is basically similar. The energy density *U*, *U*_e_, and *U*_d_ generally increased first and then decreased. In the pre-peak stage, *U*, *U*_e_, and *U*_d_ increased with the increase in axial strain. Some of the energy input from external forces is stored in the rock as elastic energy, and the other dissipated energy is used for the generation, development, and evolution of micro-fractures, intergranular friction, or resistance to CFRP constraints. Taking the reinforced area A = 3140 mm^2^ as an example (Figure 7c), we can draw the following conclusions:

(1) Before the axial force is loaded to the axial strain of 0.1975%, the energy increases slowly, the dissipated energy is slightly greater than the elastic energy, and the two energy values are at a low level of approximately 10 KJ/m^3^. The reason for this is that the original crack is compressed and leads to plastic deformation at this stage. Due to the small initial stiffness, the energy transformation is slow. After compaction, the sample enters the elastic stage, in which elastic deformation mainly occurs. In this case, energy accumulation is the main factor, and the growth rate of elastic energy is much higher than that of dissipated energy.

(2) When the axial strain exceeds 0.6923%, the growth rate of elastic energy slows down, whereas that of dissipated energy increases, which can be explained as the sample entering the volume expansion stage. The plastic deformation increases significantly and requires a lot of energy consumption. When the stress reaches the peak value of 119.83 MPa, the elastic energy also reaches the peak value of 292.6 KJ/m^3^, the energy storage of the specimen reaches the limit, and dilatancy failure occurs. The elastic energy stored in the specimen is rapidly released and converted into dissipated energy for damage and failure.

(3) In the first cycle after the peak, a large number of fractures spread to form a fracture surface, and the release of elastic energy decreases rapidly. The total energy reaches a peak of 625.03 KJ/m^3^, and the dissipated energy reaches a peak of 573.05 KJ/m^3^. In the residual stage, the total energy, elastic energy, and dissipated energy all fluctuate slightly and gradually become stable, and a certain residual strength is detected.

### 4.3. Analysis of Energy Evolution of Various Reinforcement Schemes

In order to analyze the strengthening mechanism of CFRP, taking the confining pressure σ_3_ = 15 MPa as an example, energy–strain curves, elastic energy–strain curves, and dissipated energy–strain curves of different strengthening schemes are drawn in Figure 10.

During the process from loading to failure, the total energy, elastic energy, and dissipated energy of the reinforced sandstone are all greater than those without reinforcement, and the growth rate gradually increases with the increase in the reinforced area and axial strain, reaching the peak value when the specimen fails. It can be interpreted that the CFRP restricts axial strain expansion by constraining the radial strain. If the axial strain is the same as that of the specimen without reinforcement, more energy is needed to overcome the constraint of the CFRP. When plastic failure occurs near the peak value, the axial strain increases greatly, and the energy dissipated by the CFRP reaches the peak value. Due to the constraint of CFRP, the occurrence and expansion of internal micro-cracks are limited, and more elastic energy can be accumulated in the reinforced specimens.

When the unreinforced sandstone specimen enters the expansion stage, the growth rate of elastic energy slows down, and the dissipated energy increases rapidly. However, the reinforced specimen was still in the elastic stage, and the growth rate of elastic energy was much higher than that of dissipated energy, indicating that the carbon fiber cloth effectively improved the strength of the specimen. For the residual strength stage, the axial strain, total energy, elastic energy, dissipated energy, and residual strength of the specimen all increased with the increase in the reinforced area, indicating that the CFRP effectively limited the energy release or dissipation during the fracture of the rock sample, and most of the energy was used for internal fracture failure.

### 4.4. Analysis of Energy Evolution under Various Confining Pressures

In order to analyze the influence of confining pressure on the energy evolution of reinforced sandstone, energy density curves of different confining pressures are drawn in Figure 11, Figure 12 and Figure 13.

In Figure 11, Figure 12 and Figure 13, the total energy and elastic energy of different reinforced samples in the pre-peak loading stage increase linearly, and the growth rate increases with the increase in confining pressure, indicating that confining pressure can improve the energy absorption efficiency and the energy storage limit. This is mainly due to the accumulation of elastic energy before the expansion stress is reached. The dissipated energy increases exponentially with the increase in axial strain, increases sharply in the plastic failure stage, and reaches the maximum value in the post-peak stage. With the increase in confining pressure, the maximum dissipated energy and axial strain are also larger, indicating that the higher the confining pressure, the slower the energy dissipation rate, but the more energy dissipated.

### 4.5. Analysis of Energy Distribution

In a closed system, the total input energy is converted into elastic energy and dissipated energy without considering energy exchange and frictional heat generation. During the loading process, the proportion of elastic energy and dissipated energy will affect the change of the failure mode and mechanical parameters [22]. According to the test results, the energy proportion evolution curves of different reinforcement schemes were drawn when the confining pressure was 15 MPa, as shown in Figure 14.

In the initial loading cycle, plastic deformation is the main deformation, and energy consumption is mainly the compression of cracks and pores. In the elastic stage, the proportion of elastic energy increases rapidly, and the proportion of dissipated energy is relatively small due to the small plastic deformation. In the fracture expansion stage, the growth rate of dissipated energy increases, and a large plastic deformation occurs. Although the growth rate of elastic energy slows down, it still accumulates and reaches the maximum value.

In the pre-peak loading process, the proportion of elastic energy increases with the increase in the reinforced area, indicating that CFRP can improve the energy absorption efficiency, improve the energy storage limit, inhibit the development of cracks, slow down the dissipation efficiency, and improve the peak strain of dissipated energy. When entering the post-peak stage, the accumulated elastic energy is released rapidly, a large number of cracks form connected cracks, and the dissipated energy increases sharply. However, with the increase in reinforcement area, the maximum dissipated energy ratio decreases. In the residual strength stage, the ratio of elastic energy to dissipated energy fluctuates slightly, but gradually tends to be stable, because no new fault surface is generated.

### 4.6. Analysis of Energy Damage Characteristics

The process of rock deformation and failure is essentially a process of energy transformation and energy dissipation. Energy dissipation runs through the whole process from intact rock to loading failure and instability, that is, pore compaction, new fracture generation, and expansion to the fracture surface. Therefore, energy dissipation reflects the degree of damage and deterioration of the rock. Based on the energy dissipation theory, the ratio between the dissipated energy at any time and the energy dissipation value when the strength is lost is defined as the damage variable *D* [22], whose expression is as follows:(3)$$D=\frac{U^{d}}{U_{c}^{d}}$$
where *U^d^* is the dissipated energy at any time, KJ·m^−3^; and *U^d^_c_* is the dissipated energy when the intensity is lost, KJ·m^−3^

As the formula shows, when *U^d^* is greater than *U^d^_c_*, that is, when the damage variable *D* > 1, the rock has been damaged and has lost stability, and the damage variable has no significance at this time. Therefore, this study only considers the damage evolution of the cyclic loading and unloading process before rock strength loss.

According to the test results, the evolution curve of damage variables with axial strain under different confining pressures was drawn, as shown in Figure 15. There are three stages in the rock deformation: crack closing, crack initiation, and crack damage [23]. At the initial stage of loading, the damage is mainly caused by micro-crack compaction and intergranular dislocation. In the elastic stage, the crack propagation rate is slow, and the damage variable is small. However, the damage variable increases rapidly in the stage of fracture propagation, indicating that the damage degree of the rock surges at this stage. Under the same confining pressure, with the increase in reinforcement area, the damage under the same axial strain decreases, and the maximum axial strain increases. This shows that CFRP effectively inhibits the expansion of cracks in the rock, improves mechanical properties, and reduces rock damage.

In order to analyze the influence of confining pressure on the damage variable of reinforced sandstone, the damage variable-axial strain under different confining pressure conditions was fitted (Figure 16). The correlation coefficients of the fitting curves are all greater than 0.95, and the axial strain is an exponential function of the damage variable. For the same axial strain, the larger the confining pressure, the smaller the damage variable, and the larger the axial strain at the peak stress. The reason can be explained as follows: the increase in confining pressure increases the stiffness and elastic modulus, which makes it difficult for the fracture propagation inside the rock and reduces the energy dissipation, thus reducing the damage degree of the rock.

## 5. Conclusions

By comparing the mechanical properties and energy accumulation and dissipation characteristics of rock samples with various reinforcement schemes under triaxial cyclic loading and unloading, the application of CFRP material in sandstone reinforcement is verified, and the mechanism of CFRP reinforcement is revealed from the perspective of energy. The main conclusions are as follows:In the triaxial cyclic loading and unloading test, the reinforcement effect of the CFRP material was relatively ideal, and the initial damage caused by bolt drilling was effectively avoided, which provides a new direction for the research on underground engineering support materials.CFRP improved the strength deformation parameters and shear strength parameters of sandstone effectively, and the reinforcement effect increased with the increase in the reinforced area.CFRP restricted the axial strain expansion by limiting the radial strain and inhibited the occurrence and expansion of internal microcracks so that more elastic energy can be accumulated during loading. This was the intrinsic reason for improving the strength of the specimen.CFRP began to play a role in the early stage of loading, and the reinforcement effect became increasingly obvious as the loading entered the plastic deformation stage. In the residual strength stage, the CFRP effectively limited the energy release or dissipation during rock failure.As the reinforcement area increased, the energy characteristic density and peak strain increased. However, the proportion of elastic energy increased, and the proportion of dissipated energy decreased, indicating that the CFRP improved the energy absorption efficiency, enhanced the energy storage limit, slowed down the dissipation efficiency, and increased the peak strain of dissipated energy.Under the same confining pressure, the damage variable decreased with the increase in the reinforcement area when the same coaxial strain occurred. Therefore, CFRP annular reinforcement reduced intergranular dislocation and inhibited the development of cracks and damage in the rock mass, which is the essence of constraint reinforcement.

## Figures and Tables

**Figure 1 materials-16-00211-f001:**
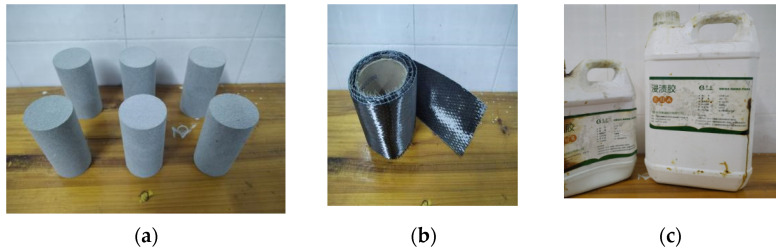
Experiment material: (**a**) sandstone specimen; (**b**) CFRP; (**c**) impregnated glue.

**Figure 2 materials-16-00211-f002:**
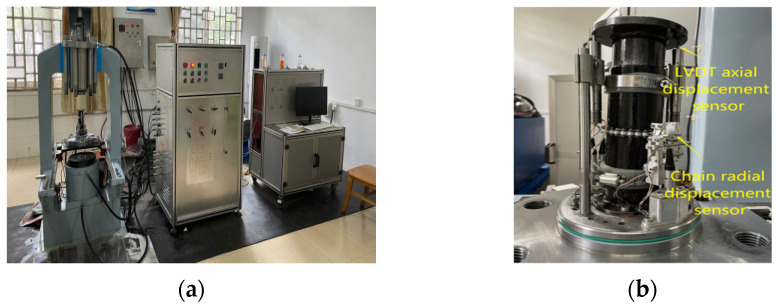
Test equipment and rock specimens: (**a**) automatic rock triaxial test system; (**b**) sensor.

**Figure 3 materials-16-00211-f003:**
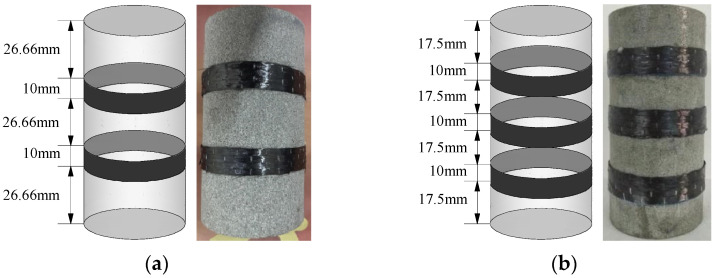
Sandstone with different circumferential reinforcement schemes: (**a**) scheme 1: A = 3140 mm^2^; (**b**) scheme 2: A = 4710 mm^2^.

**Figure 4 materials-16-00211-f004:**
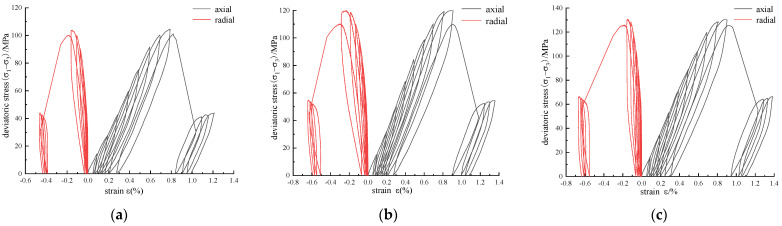
Stress–strain curve of triaxial cyclic loading and unloading with *σ*_3_ = 15 MPa: (**a**) unreinforced: A = 0 mm^2^; (**b**) scheme 1; (**c**) scheme 2.

**Figure 5 materials-16-00211-f005:**
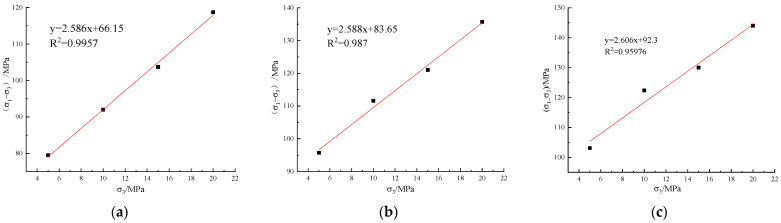
Relationship between peak strength and confining pressure under cyclic loading and unloading: (**a**) unreinforced; (**b**) scheme 1; (**c**) scheme 2.

**Figure 6 materials-16-00211-f006:**
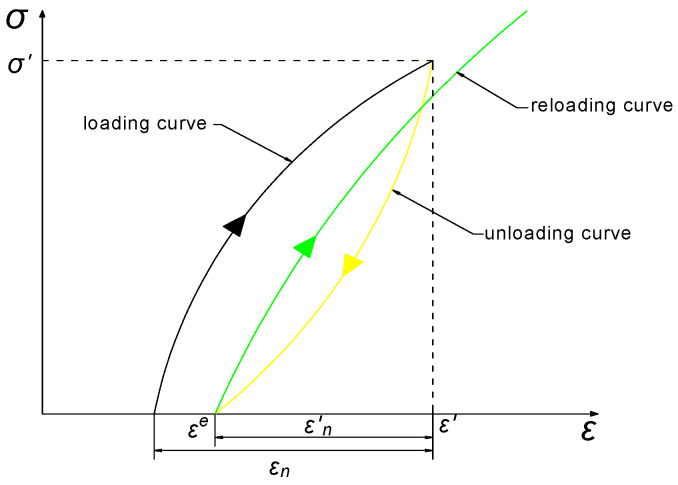
Schematic of energy density calculation for cyclic loading and unloading.

**Figure 7 materials-16-00211-f007:**
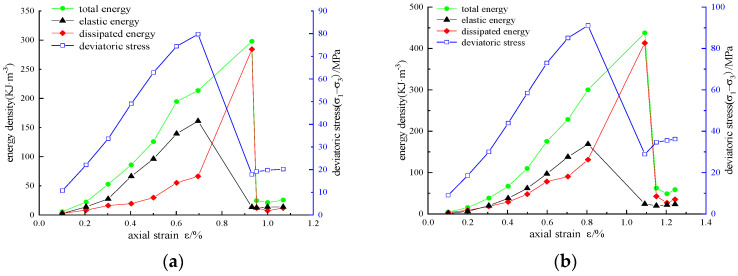
Variation curves of stress, strain, and energy density of unreinforced samples: (**a**) *σ*_3_ = 5 MPa; (**b**) *σ*_3_ = 10 MPa; (**c**) *σ*_3_ = 15 MPa; (**d**) *σ*_3_ = 20 MPa.

**Figure 8 materials-16-00211-f008:**
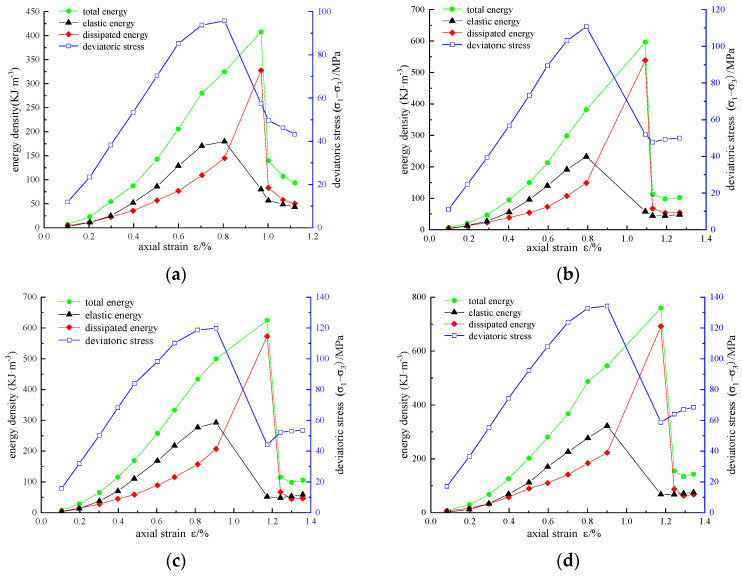
Variation curves of stress, strain, and energy density under Scheme 1: (**a**) *σ*_3_ = 5 MPa; (**b**) *σ*_3_ = 10 MPa; (**c**) *σ*_3_ = 15 MPa; (**d**) *σ*_3_ = 20 MPa.

**Figure 9 materials-16-00211-f009:**
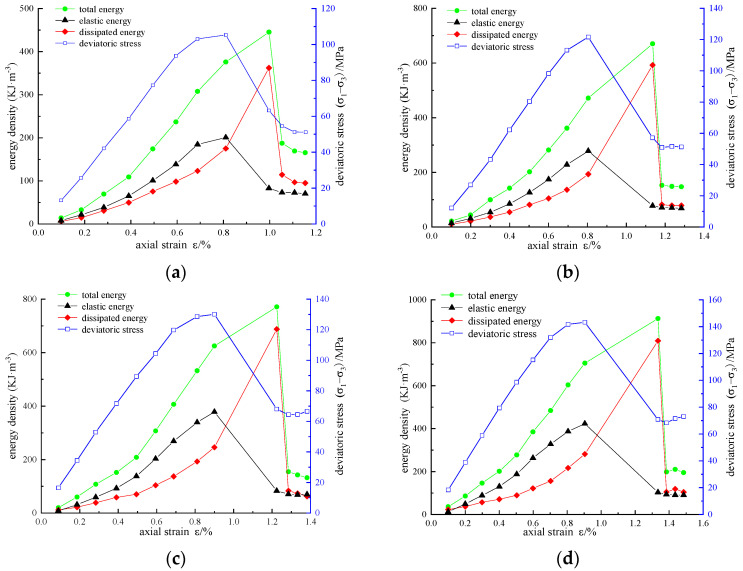
Variation curves of stress, strain, and energy density under Scheme 2: (**a**) *σ*_3_ = 5 MPa; (b) *σ*_3_ = 10 MPa; (**c**) *σ*_3_ = 15 MPa; (**d**) *σ*_3_ = 20 MPa.

**Figure 10 materials-16-00211-f010:**
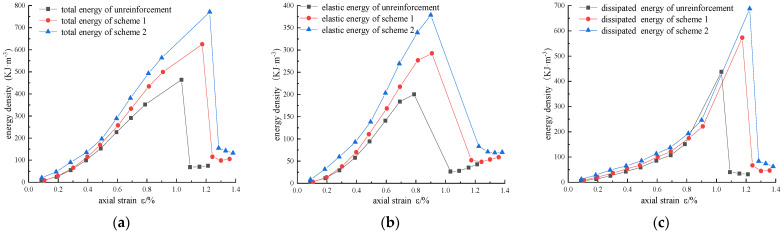
Comparison of energy density variation with strain under different reinforcement schemes: (**a**) total energy; (**b**) elastic energy; (**c**) dissipated energy.

**Figure 11 materials-16-00211-f011:**
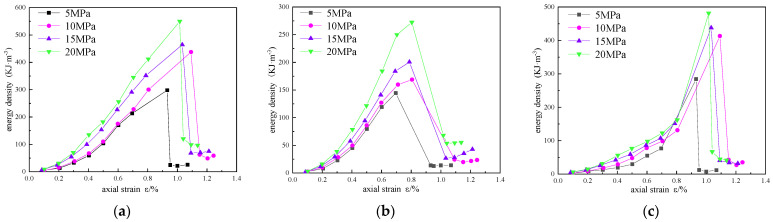
Energy evolution curves of unreinforced sandstone under different confining pressures: (**a**) total energy; (**b**) elastic energy; (**c**) dissipated energy.

**Figure 12 materials-16-00211-f012:**
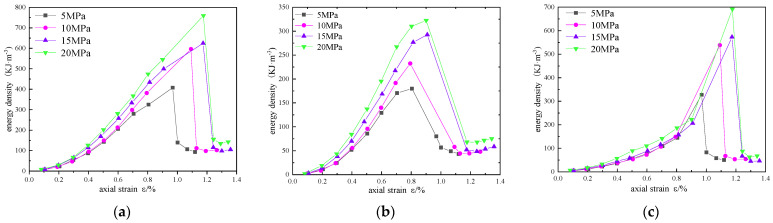
Energy evolution curves of scheme 1 sandstone under different confining pressures: (**a**) total energy; (**b**) elastic energy; (**c**) dissipated energy.

**Figure 13 materials-16-00211-f013:**
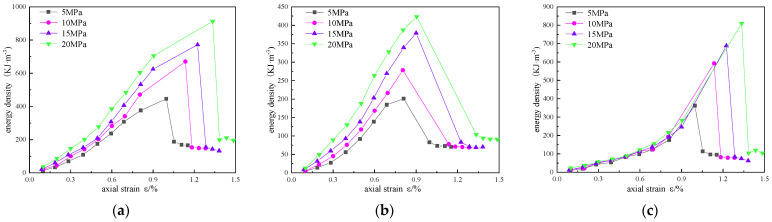
Energy evolution curves of scheme 2 sandstone under different confining pressures: (**a**) total energy; (**b**) elastic energy; (**c**) dissipated energy.

**Figure 14 materials-16-00211-f014:**
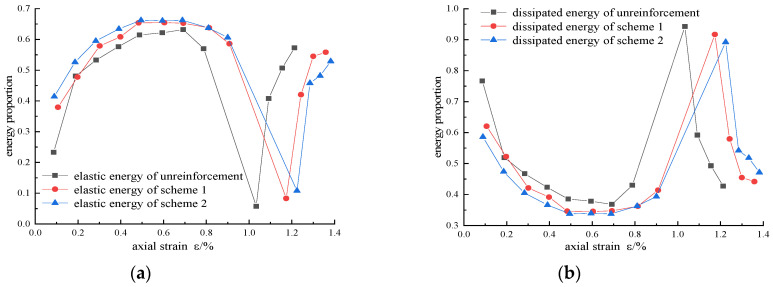
Energy proportion evolution curves of various reinforcement schemes when σ_3_ = 15 MPa: (**a**) elastic energy; (**b**) dissipated energy.

**Figure 15 materials-16-00211-f015:**
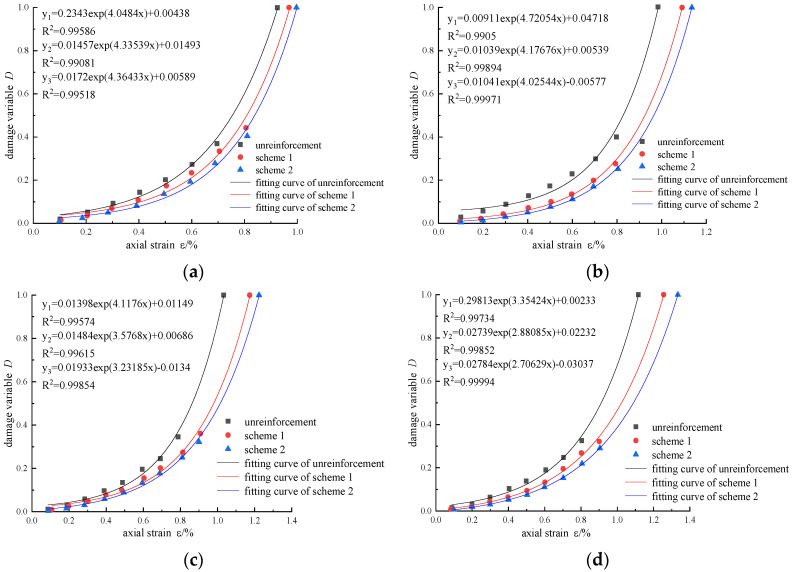
Variation in damage variable with axial strain: (**a**) *σ*_3_ = 5 MPa; (**b**) *σ*_3_ = 10 MPa; (**c**) *σ*_3_ = 15 MPa; (**d**) *σ*_3_ = 20 MPa.

**Figure 16 materials-16-00211-f016:**
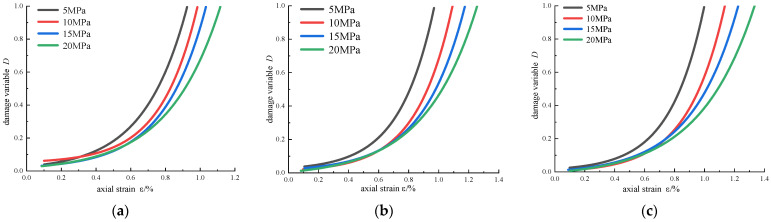
Comparison curves of damage variables under different confining pressures: (**a**) unreinforced; (**b**) scheme 1; (**c**) scheme 2.

**Table 1 materials-16-00211-t001:** Technical parameters and indexes of anchorage materials.

Parameters	Tensile Strength (MPa)	Elasticity Modulus (Gpa)	Elongation (%)	Bending Strength (MPa)
CFRP	≥3000	≥200	≥1.5	≥600
Impregnated glue	≥35	≥2200	≥1.3	≥40

**Table 2 materials-16-00211-t002:** Results of triaxial cyclic loading and unloading tests under various restraint schemes with σ_3_ = 15 MPa.

Reinforcement Schemes	Peak Strength (MPa)	Axial Peak Strain (%)	Radial Peak Strain (%)	Residual Strength (MPa)	Cohesion (MPa)	Friction Angle (°)
Unreinforced	104.61	0.7875	−0.1589	42.05	20.57	26.25
Scheme 1	119.94	0.9097	−0.2071	52.29	26	26.27
Scheme 2	130.19	0.9248	−0.2747	64.41	28.05	27.41

## Data Availability

The data presented in this study are available on request from the corresponding author. The data are not publicly available due to privacy.

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
