# Peer review of "Energy Action Mechanism of Reinforced Sandstone under Triaxial Cyclic Loading and Unloading"

_materials, 2022, doi:10.3390/ma16010211_

Round 1

Reviewer 1 Report

This work is certainly interesting and has important potential consequences. But the manuscript needs the following minor modifications:

1.      Page 4, Section 2.3. Test Scheme: The units of the loading rate written as “Mpa/s”, it must be corrected to “MPa/s”.

2.      Page 5, 2nd Paragraph: Authors are recommended to use superscripts at “The area of the hysteresis loop can be represented as the area enclosed by the nth unloading curve and the nth+1 loading curve …”

Author Response

We have modified it according to your suggestions. Please see the reply document. Thank you very much.

Reviewer 2 Report

The paper is well written, but it is of average quality. It presents a large amount of experimental work. Also, the evaluation and the comparisons of the results are well done. However, it is necessary to extend the conclusions and literature by comparing the scientific results with foreign research.

Author Response

(The authors gave the same response as above.)

Reviewer 3 Report

Dear Authors,

The manuscript explained the energy action mechanism of reinforced sandstone under triaxial cyclic loading and unloading, which is organized well; however the following modification would be needed to match the journal criteria:

1) The introduction is very general from a special viewpoint on the paper topic. The authors are highly encouraged to add more specific relevant recent papers to this section.

2) Section 2.1: Please clarify/add how the authors get a sample from 800m bgl.

3) Section 2.1: Please add any possibilities to measure the suction/pore water pressure on the equipment set.

4) Section 3.1: Why only the results for confining pressure 15MPa is shown in Figure 4?

5) Table 2: The internal friction angle between unreinforcement and schemes 1  and 2 has no significant change; however, the cohesion is more sensitive. The author should add an explanation of how it was obtained.

6) section 4.2: The energy evolution, which is the main of this paper, is not deeply discussed scientifically. It shows the graph from the test; however, a quantitative and qualitative explanation is missed.

7) conclusion should be revised based on previous comments.

8) The paper needs to be checked by a native English technical staff.

Author Response

(The authors gave the same response as above.)

Reviewer 4 Report

Authors have determined the bearing capacity of rock mass supported by bolt and shotcrete and simulated by the circular reinforcement method of CFRP. Following are the comments

1. The abstract should be rewritten and should be more quantitative

2. Author should cite and discuss the similar work done by researchers outside china also like

Temperature-dependent thermophysical properties of Ganurgarh shales from Bhander group, India, AK Verma, MK Jha, S Maheshwar, TN Singh… - Environmental Earth Sciences, 2016

Damage characteristics of jalore granitic rocks after thermal cycling effect for nuclear waste repository, PK Gautam, R Dwivedi, A Kumar, A Kumar, AK Verma, KH Singh, Rock Mechanics and Rock Engineering 54 (1), 235-254

3D instability analysis of an underground geological repository—an Indian case study, AK Verma, RK Bajpai, TN Singh, PK Narayan, A Dutt - Arabian Journal of Geosciences, 2011

3. Statistical modeling part is very weak and more details of the model should be given. More correlation should be derived.

4. Study area is very weak and should include more details of the study area

6. Pease validate the obtained results with field condition

7. Please highlight the innovative component of the work

Author Response

(The authors gave the same response as above.)

Reviewer 5 Report

The manuscript should be given a full review by English editing service. Some errors complicate the ability of the reader to clearly understand what the authors intend to communicate. 

The introduction section should be more cleverly constructed, by regrouping the papers using similar approaches and summarizing their message in a single sentence. It is also important to point out the differences between previous work and the current study. In its present form, it is not easy to discern how the paper provides something new concerning previous work by other authors. It is not explained why this method was chosen in this study and why other methods in the literature were not used.

The CFRP is not mentioned at all in the introduction section but all results are related to CFRP. How important the CFRP is for this study should be explained.

The whole paper should be reorganized. For example, the methodology should not be given after the results. Another major issue is too many titles in the article. Detailing all the technical issues encountered and how they were solved. The other important issue is that it is not easy for the reader to understand the novelty of the approach adopted by the authors.

The aim and results of this study are not understood (energy characteristics of the reinforced rock or CFRP role). What are the intended and found results of this study? The results should be discussed considering previous studies.

Author Response

(The authors gave the same response as above.)

Reviewer 6 Report

The reviewed article deals with the topic: " Energy action mechanism of reinforced sandstone under triaxial cyclic loading and unloading". This is a very interesting issue, which is suitable for a detailed research. The authors offer in this article an original view on the investigated theme.

From the professional point of view, I consider the presented information to be new and original. The article has a scientific level and but its structure is not appropriately chosen. With regard to the professional content, this article fully fits into the concept of the given journal. Processing of the literature sources is sufficiently performed.

From the theoretical point of view, the investigated problems are clearly and comprehensibly described. The presented results are very interesting and they create a solid starting base for further scientific research. At the same time, I am convinced about direct applicability of the obtained results in the practice. Globally, I have the following comments to this article:

1. The chapter Conclusion needs to be significantly expanded and reworked, whereby it is necessary to emphasize the obtained results, the conclusions and orientation of a further development.

2. The structure of the paper should be significantly reworked. I believe that these my comments will help the authors to improve their article. The structure of the paper should be :

Article structure

Introduction

Material and methods
Provide sufficient details to allow the work to be reproduced by an independent researcher. Methods that are already published should be summarized, and indicated by a reference. If quoting directly from a previously published method, use quotation marks and also cite the source. Any modifications to existing methods should also be described.

Theory/calculation
A Theory section should extend, not repeat, the background to the article already dealt with in the Introduction and lay the foundation for further work. In contrast, a Calculation section represents a practical development from a theoretical basis.

Results
Results should be clear and concise.

Conclusions
The main conclusions of the study may be presented in a short Conclusions section, which may stand alone or form a subsection of a Discussion or Results and Discussion section.

Author Response

(The authors gave the same response as above.)

Round 2

Reviewer 5 Report

 Accept in present form

Reviewer 6 Report

Accept in present form